# Evaluation of Park Accessibility Based on Improved Gaussian Two-Step Floating Catchment Area Method: A Case Study of Xi'an City

**Yunmei Li [1,*], Yuanli Xie [1,2,*], Shaoqi Sun [1] and Lifa Hu [1]**

[1]   College of Urban and Environmental Sciences, Northwest University, Xi'an 710127, China; sean20210511@gmail.com (S.S.); hlf_2019@163.com (L.H.)
[2]   Shaanxi Key Laboratory of Earth Surface System and Environmental Carrying Capacity, Northwest University, Xi'an 710127, China
[*]   Correspondence: liyunmei2020@163.com (Y.L.); xieyuanli@126.com (Y.X.)

**Abstract:** Park accessibility plays a critical role in evaluating the quality of park construction. However, the conventional accessibility model ignores non-spatial factors, so it is crucial to use more complex methods for evaluating park accessibility. This study aims to establish an improved Gaussian-based two-step floating catchment area method (iG2SFCA) based on Point of Interest (POI), population data and Baidu map, to measure the park accessibility of various travel modes (walking, riding and driving modes) in 5-min, 15-min and 30-min scenarios, and location quotient is used to assess spatial equity of parks. The results show that: (1) There are clear disparities between park supply and population demand at the street level. (2) iG2SFCA evaluates the level and attractiveness of the park comprehensively. It is more sensitive to identifying accessibility, which can lead to a more realistic assessment of Park accessibility. (3) Under the three modes of transportation, the accessible area of the park increases with time, and the accessibility difference between residential areas is the smallest under the 30-min scenario. Overall, accessibility of park is relatively high; however, there is an obvious tendency for the accessibility level to decrease from the park as the center. The areas with poor accessibility appear in the north and southeast of the research area. (4) There are regional variances in the spatial equity of parks within Xi'an 3 City, and the park configuration needs to be optimized. These findings can provide theoretical support for further optimizing the layout of park in Xi'an in order to improve the spatial equity of urban parks.

**Keywords:** park; accessibility; Gaussian based two-step floating catchment area (2SFCA) method; big data; Xi'an

## 1. Introduction

As an important component of ecological products, urban parks serve not only ecological functions but also social functions, which can help cities enhance their environmental quality [1]. Numerous studies have proven that parks can improve physical health, reduce psychological stress [2–4], boost communication among residents [5], and improve residents' well-being [6]. In addition, as a third space, parks can meet the daily needs of inhabitants' for leisure activities, and positively promote the harmonious development of the urban social environment.

Park accessibility, a critical criterion for determining whether the layout of a park is balanced, is significant to the close integration of ecological civilization construction and urban development. Accessibility was originally presented as a notion in transportation geography, defined as the extent to which two nodes in a transportation network can communicate [7], and was later introduced into human geography [8]. Park accessibility refers to the proximity of a residential area to a park, in other words, residents' capacity to overcome travel expenses (time and distance) in order to visit the park [9].

A number of studies have been conducted on measurement methods of park accessibility. Early, measurement techniques included the ratio methods [10] and buffer analysis methods [8]. The computation approach is simple and intuitive, as well as easy to implement in urban planning. The widespread application of GIS encourages the growth of the cost weighted distance model [11], the minimal nearest neighbor distance method [12], and the network analysis method [13,14]. By establishing a particular resistance value, this approach may represent the cost of reaching the park from various destinations depending on the actual road network and can accurately portray the park's accessibility, assuming the essential data are complete and the computational capability permits. Due to the fact that the potential model and the two-step floating catchment area (2SFCA) take into account the supply capacity of the park and the demand of residents, it has become a widely used calculation method. However, there are still some issues in the 2SFCA: while using the binary division approach to determine the search range according to the threshold, the search beyond the domain is fully inaccessible; all supply points in the search domain have the same attraction; ignoring the distance attenuation influence on demand points [15]. A number of improved models have been proposed in response to these flaws: kernel density two-step moving search method (K2SFCA) [16], variable width search method (VFCA) [17], three-step floating catchment area method (3SFVA) [18], and G2SFCA [19]. Among them, the G2SFCA is a more scientific accessibility measurement method by introducing the Gauss attenuation function to fit the changing relationship between park attractiveness and distance, which conforms to the travel characteristics of residents and considers the supply scale and population demand of the park. However, it did not take into account non-spatial factors. Therefore, to overcome shortcomings in classic accessibility models, several scholars incorporated new criteria gauging a park's attractiveness. Dony [20] assessed park accessibility by looking at the park size and on-site amenities. Xing [21] incorporated park size and function into the 2SFCA model in order to provide a more accurate assessment of park accessibility. According to several surveys, even if it is not the closest park, residents choose high-quality parks under the influence of non-spatial factors [22–26]. However, previous research did not adequately consider the park's quality, research that incorporates park level and park attractions into accessibility models is rare.

Most previous studies depend on census data at a sub-district level [27–29], or divide the research region into grids and distribute the population to each grid evenly. In recent years, the use of big data to modify traditional processed data has emerged as a new trend in geographic research [30], allowing for a finer scale of research. Guo [31,32] uses mobile phone signaling data to estimate area population density, but the cost of acquiring data is too high. The data on the number of households by the Anjuke platform can reflect the static population distribution and the precise demand for parks. In addition, the majority of traffic trip data are derived from network analysis results, but it is difficult to collect complete basic data. At the moment, major map providers can improve route time predictions for multiple means of transport, which has been establish by relevant research as accurate and reliable [33,34].

Therefore, Xi'an was chosen as a research area, and the iG2SFCA was proposed in this study using the Baidu map route planning interface based on the POI and population data. This paper aims to: (1) Compare two methods before and after improvement to test the practicability of the iG2SFCA (2) explore the accessibility of the park in Xi'an in 5-min, 15-min, and 30-min scenarios, under three trip modes (walking, cycling, and driving mode); (3) study the spatial equality of park distribution in Xi'an. This paper is structured as follows: The following section introduces the study area and data preparation. The third part introduces the G2SFCA and explains the iG2SFCA. The fourth section illustrates the assessment results. Specifically, the comparison between two methods and the accessibility features in various modes. The final section summarizes and discusses the findings of this research.

## 2. Materials and Methods

### 2.1. Study Area

Xi'an, the only mega city in northwest China, is the capital city of Shaanxi province. It possesses distinct historical and cultural genes and is home to numerous heritage parks. The city was named "National Forest City" in 2016. By 2020, the permanent population had reached almost 12.96 million, with 7.61 million urban residents. There are 117 parks in all, covering a total area of 3490.53 hm$^2$. The built-up area has a green coverage rate of 39.32%. This study focuses on the area within the Xi'an City Ring Expressway, which encompasses six districts in the main city that serves as center area of the permanent population distribution.

### 2.2. Data and Preprocessing

In accordance with the definition in "Park Design Specification", the park data and the area of interest (AOI) in the Amap were collected and trimmed based on the 2 m image of Xi'an in 2019. There are 79 parks in all within the research area. Due to the length and narrowness of the Tang City Wall Heritage Park and the Ring Park, they were separated into sections in line with the main road, and serial numbers were applied to distinguish them so as to facilitate subsequent research. Ultimately, a total of 100 parks were recognized. According to the City Park Classification Standard DBJ61/T110-2015 and the green space system planning reports of Xi'an by urban planning department parks are divided into comprehensive park, theme park and community park, considering the grade, scale and facilities of parks. The spatial distribution of various parks is shown in Figure 1.

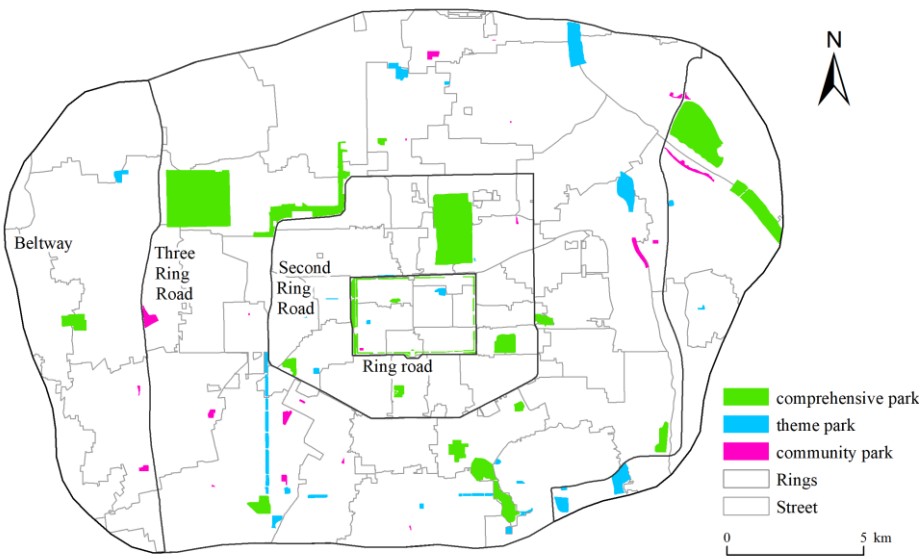

**Figure 1.** Distribution of parks within Xi'an Ring Expressway.

This project utilizes the Anjuke platform and python to collect residential district data within Xi'an City Ring Expressway. After data collection, 501 residential districts with no household data were deleted, with 4701 residential neighborhoods remaining as valid data. According to the 2020 statistical data of Xi'an (http://tjj.xa.gov.cn/tjnj/20 20/zk/indexch.htm, accessed on 10 August 2021), the average permanent population of each household in Xi'an is 3.0, hence the total population is calculated by multiplying the number of households in the community by 3.0.

Due to the fact that different demographic groups favored different modes of transport [21], we analyzed the accessibility of the park using the three modes of transport—walking, cycling, and driving. Among them, the transit time from the residential area to the park is obtained entirely through the Baidu Map API interface [35], which involves taking the residential area as the starting point and the park as the destination, constructing the OD matrix, and

then using python to request Baidu Maps' lightweight route planning service to obtain the transit time. This study recorded travel time in a week from 5 July 2021 to 11 July 2021. The travel time was gathered daily at 18:00, and the average time of seven days was used to determine the actual time taken from the residential area to the park.

*2.3. Methods*

2.3.1. G2SFCA Method

Dai [19] presented the 2SFCA approach for evaluating the accessibility of green space in Atlanta, Georgia, the United States. The fundamental calculating idea is divided into two steps:

Step 1: Calculate the supply–demand ratio. Determine a distance threshold for each park in the research area, calculate the demand within the threshold range, and multiply it by the Gaussian function, and divide the park supply by the result of multiplication to obtain the supply–demand ratio.

Step 2: Calculate accessibility. For each demand point, search all parks within the threshold range, multiply the park's supply–demand ratio by the Gaussian function, accumulate the results, and obtain the accessibility $A_i$ of each demand point.

$$A_i = \sum_{j \in \{d_{kj} \le d_0\}} G(d_{kj}, d_0) R_j = \frac{S_j}{\sum_{k \in \{d_{kj} \le d_0\}} G\left(d_{kj}, d_0\right) P_k} \tag{1}$$

$$G\left(d_{kj}, d_0\right) = \begin{cases} \dfrac{e^{-\left(\frac{1}{2}\right) \times \left(\frac{d_{kj}}{d_0}\right)^2} - e^{-\left(\frac{1}{2}\right)}}{1 - e^{-\left(\frac{1}{2}\right)}}, & if \quad d_{kj} \le d_0 \\ 0, & if \quad d_{kj} > d_0 \end{cases} \tag{2}$$

In the formula: $S_j$ is the supply of the park $j$, $d_0$ denotes the distance threshold, $d_{kj}$ is the distance between the supply and the demand points, and $P_k$ denotes the demand within the threshold range, which is frequently stated in terms of population. $G(d_{kj}, d_0)$ is the Gaussian attenuation function.

2.3.2. iG2SFCA Method

Although the G2SFCA assesses park accessibility from both supply and demand perspectives, there are still deficiencies. This paper is enhanced in two aspects as follows.

Firstly, consider the contrasts in park quality and park attractiveness factors. In addition to spatial features, park size, and nearby facilities also have a significant effect on park accessibility [10,36,37]. As a result, we assign attraction weights of 0.6, 0.4, and 0.2 to the three levels of classification. With an appropriate walking distance of 800 m [38], we take the normalized quantity of POI within the 800 m buffer outside the park as an indicator of attractiveness. The normalized quantity is calculated as the attractiveness factor. This article focuses on five types of POIs that are closely related to leisure and entertainment [39]: catering services, shopping services, sports and leisure services, scenic places, and scientific, educational, and cultural services; with a total of 77,089 POIs retrieved.

Secondly, the distance threshold is replaced with the time threshold. Due to the advancement of big data technology, it is possible to utilize a Baidu map to acquire travel time under multiple travel modes, which facilitates the analysis of the park's spatial accessibility and reveals the accessibility variations under different traffic modes [10]. Consequently, this study used three modes of transportation—walking, riding, and driving—to examine the accessibility change characteristics at various time thresholds.

The revised calculating formulas are as follows:

$$A_{ij} = \sum_{j \in \{t_{kj} \le t_0\}} G(t_{kj}, t_0) R_j = \frac{W_j S_j}{\sum_{k \in \{t_{kj} \le t_0\}} G\left(t_{kj}, t_0\right) P_k} \tag{3}$$

$$W_j = N_j \times A_j \tag{4}$$

$$G\left(t_{kj}, t_0\right) = \begin{cases} \dfrac{e^{-(\frac{1}{2}) \times (\frac{t_{kj}}{t_0})^2} - e^{-(\frac{1}{2})}}{1 - e^{-(\frac{1}{2})}}, & if \quad t_{kj} \le t_0 \\ 0, & if \quad t_{kj} > t_0 \end{cases} \tag{5}$$

In the formula, $A_{ij}$ represents the accessibility of park $j$ under traffic mode $i$, $R_j$ represents the park's supply–demand ratio, $S_j$ represents the area of park $j$, $W_j$ represents the attractiveness of park $j$, expressed as the product of the normalized index $N_j$ of the number of POIs within an 800 m buffer and the park's type weight $A_j$, $P_k$ represents the population, $t_0$ represents the time threshold, $t_{kj}$ represents the transit time between the park and the community, and $G(t_{kj}, t_0)$ is the Gaussian attenuation function.

2.3.3. Location Quotient

The location quotient is the ratio of the park area enjoyed by a street's population to the per capita park area in the research area, which can indicate the equity of street parks distribution [40]. Typically, the location quotient is compared to 1. When it is greater than 1, it implies that the street's park distribution level is greater than the average value of the study area. The greater the location quotient is, the greater the degree of street park distribution level is. The calculation formula for location quotient is:

$$LQ_i = \left(\frac{A_i}{P_i}\right) / \left(\frac{A}{P}\right) \tag{6}$$

In the formula, $LQ_i$ denotes the location quotient of street $i$, $A_i$ and $P_i$ denotes the park area and population of the street $i$ respectively, and $A$, $P$ respectively represent the total area and population of the park in the study area.

**3. Results**

*3.1. Supply Analysis*

Because some streets are divided by the ring expressway, this article only counts streets with an area of more than 50% in the ring expressway. There are 49 streets within the ring expressway, and parks are unevenly distributed among street (Table 1). In terms of quantity, there are nine streets with on parks situated around. Zhangbagou Street contains the largest number of parks compared to other streets, with a total of 14. This is because the Tang City Wall Heritage Park is scattered across the street in a long and narrow shape, split into many block parks by the road network. Due to the large area of Weiyang Palace Heritage Park, Daming Palace Heritage Park, and Chanba Wetland Park, the streets around these parks contain fewer number of parks, with a relatively high coverage rate of parks in contrast. Between West Third Ring Road and Ring Expressway, the parks are small and scattered due to spatial distribution.

*3.2. Demand Analysis*

Data from residential areas and data from streets within the ring expressway were connected and compared to investigate demand variances of each street (Figure 2). According to loop lines, the population is concentrated between the second and third ring roads. Meanwhile, the population within the ring road is relatively small, because of the implementation of the "Imperial City Rejuvenation" plan, which limits population growth within the ring road and adjusts the land use to reduce the height and density of buildings, transforming it into an urban functional area dominated by tourism and commerce. With the high density of firms in the high-tech zone, its rapid development has attracted an increasing number of talents. Moreover, communities in high-tech zone are primarily erected in recent years, with higher floors and larger capacities, resulting in a massive population in Yuhuazhai Street. On the other hand, the east area of the East Third Ring

Road is sparsely populated. Land of streets in northeast area is primarily for agricultural. Rural areas have a higher proportion of self-built residences, which is hard to quantify, with consequently lower population than other streets.

**Table 1.** Area table of some street Parks.

| Street Name | Number of Parks (a) | Area of Park (km²) | Area Ratio (%) | Street Name | Number of Parks (a) | Area of Park (km²) | Area Ratio (%) |
|---|---|---|---|---|---|---|---|
| Weiyanggong Street | 2 | 6.22 | 30.19 | Shilipu Street | 2 | 0.63 | 4.89 |
| Baqiao Street | 6 | 3.36 | 13.84 | Dongguannanjie Street | 1 | 0.51 | 19.47 |
| Dayanta Street | 7 | 1.75 | 20.91 | Xiguan Street | 6 | 0.42 | 8.20 |
| Daminggong Street | 1 | 1.65 | 15.68 | Dengjiapo Street | 2 | 0.40 | 2.86 |
| Zhangbagong Street | 14 | 1.21 | 4.78 | Changyanbao Street | 9 | 0.31 | 2.25 |
| Xiwang Street | 7 | 1.09 | 5.58 | Xujiawan Street | 3 | 0.30 | 3.67 |
| Ziqianglu Street | 1 | 0.95 | 39.92 | Zhangjiabao Street | 4 | 0.29 | 2.29 |
| Taihualu Street | 3 | 0.90 | 17.68 | Tanjia Street | 3 | 0.27 | 1.47 |
| Qujiang Street | 8 | 0.72 | 4.98 | Hancheng Street | 2 | 0.26 | 1.23 |
| Sanqiao Street | 4 | 0.71 | 2.23 | Huanchengxilu Street | 3 | 0.22 | 8.81 |

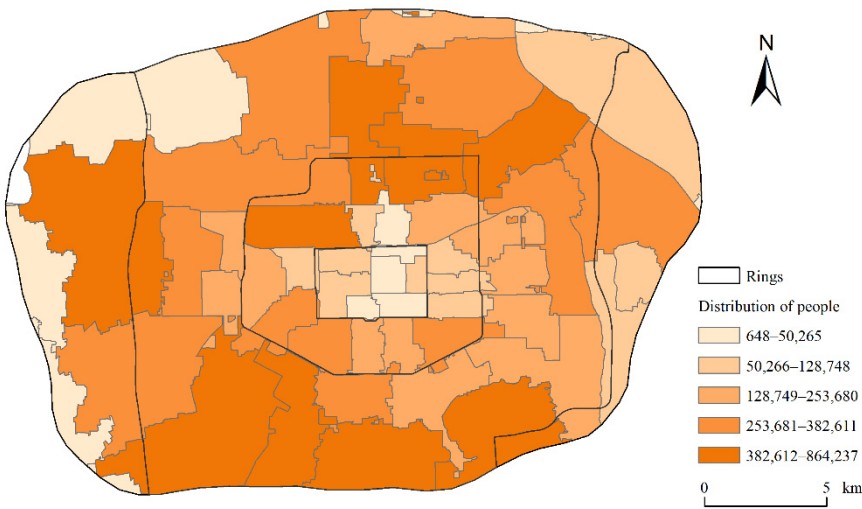

**Figure 2.** Population distribution in Xi'an Ring Expressway.

*3.3. Comparing Analysis*

Walking mode is chosen based on a 15-min threshold to compare the methods prior to and after the improvement. According to the statistical findings, significant differences between the maximum and minimum values of the G2SFCA and iG2SFCA are discovered by comparing the mean and standard deviation of the two methods reveals, which indicates that the distribution of park resources in the region is unbalanced. By comparing the mean and standard deviation of calculation results(Table 2), it is discovered that the mean value of iG2SFCA is less than that of G2SFCA. This is because the iG2SFCA considers the attractiveness of park type and the surrounding POI, and normalizes the number of POI (between 0–1), thus making the calculation result smaller.

**Table 2.** Comparison of accessibility calculation results.

| | Maximum | Minimum | Mean | SD |
|---|---|---|---|---|
| G2SFCA | 274.61 | 0.10 | 1.66 | 12.82 |
| iG2SFCA | 87.62 | 0.10 | 0.53 | 2.27 |

Normalized spatial distribution of the two approaches is shown in Figure 3. In terms of spatial distribution, the two approaches produce outcomes with a similar distribution of high values. Both methods identify that Chanba Wetland Heritage Park, Xi'an Park around

city, and Qujiang Pond Heritage Park have a high degree of spatial accessibility. However, there are some discrepancies between the two outcomes. The results of the traditional method are obviously divergent, being unable to reflect the internal accessibility differences. Calculation results of the improved method, on the other hand, are more evenly distributed and exhibit obvious transitions, which can better reflect the internal variations among regions and are more sensitive to high value identification. Such as Xingqing Palace Park and Olympic Park, the iG2SFCA takes the park level and surrounding POI into account and determines that these two parks are high accessibility areas, which is more accurate than the traditional method.

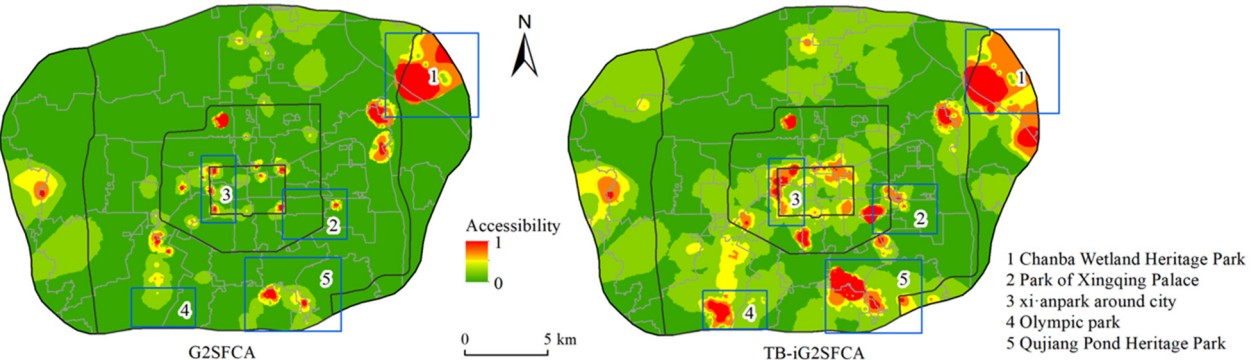

**Figure 3.** Comparison of spatial distribution of accessibility.

### 3.4. Accessibility Analysis

Given the various forms of transport, this article focuses on three frequently used travel modes: walking, cycling, and driving. The accessibility of 5-min, 15-min, and 30-min scenarios are calculated to provide a reference for inhabitants to choose their daily leisure travel mode. The mean and standard deviation of the accessibility level of each residential area for various means of transport are shown in Table 3. In the walking mode, the mean value of accessibility increases from 0.3 to 0.66 as travel time increases, and the reachable area continues to expand. In the cycling mode, the largest mean value is 0.92 in the 5-min scenario, and the SD steadily reduces from 5.60 to 0.76, demonstrating that riding is more convenient in the 5-min scenario, but the degree of convenience differs significantly across different residential communities. The mean value of residential community accessibility in driving mode is directly proportional to time, while the standard deviation is inversely proportional. When comparing the three modes of transport in a 30-min scenario, SD is the smallest at 1.99, 0.76, and 0.47. This demonstrates that as time passes, the accessibility gap between streets become smaller.

**Table 3.** Mean and standard deviation of accessibility level of residential area under different travelmodes.

| Scenarios | Walking | | Cycling | | Driving | | Multi-Mode | |
|---|---|---|---|---|---|---|---|---|
| | Mean | SD | Mean | SD | Mean | SD | Mean | SD |
| 5-min | 0.30 | 2.90 | 0.92 | 5.60 | 0.36 | 1.37 | 0.28 | 1.01 |
| 15-min | 0.53 | 2.27 | 0.68 | 1.77 | 0.58 | 1.22 | 0.32 | 0.50 |
| 30-min | 0.66 | 1.99 | 0.68 | 0.76 | 0.66 | 0.50 | 0.34 | 0.47 |

In order to compare the results from a spatial perspective, this study employs the kriging method for global analysis and categorizes the reachable areas according to the nature break to indicate the accessibility (Figure 4). Under three travel modes, the accessibility area of space expands with the growth of time. In walking mode, the area exhibits a point discrete distribution in 5-min scenario. These neighborhoods are centered along the Ring Road, near the Tang City Wall Heritage Park, and in the Qujiang neighborhood. Due to the large area of the Daming Palace, this region is inaccessible in a 15-min scenario. While in

30-min scenario, it becomes highly accessible. Under the 30-min scenario, the accessibility of most places remains 0, indicating that some residential communities are unable to use city parks effectively within 30-min. Since 18:00 is the evening rush hour in Xi'an, there may be traffic jams, and some roads sections cannot be driven when traveling by car. As a result, the riding reachable range is greater than the driving in the 5-min scenario. When the reach of the three modes of travel is compared, it is observed that walking accessibility can basically cover the area within the ring road during a 15-min scenario, while cycling and driving can effectively cover almost the entire area within the ring expressway.

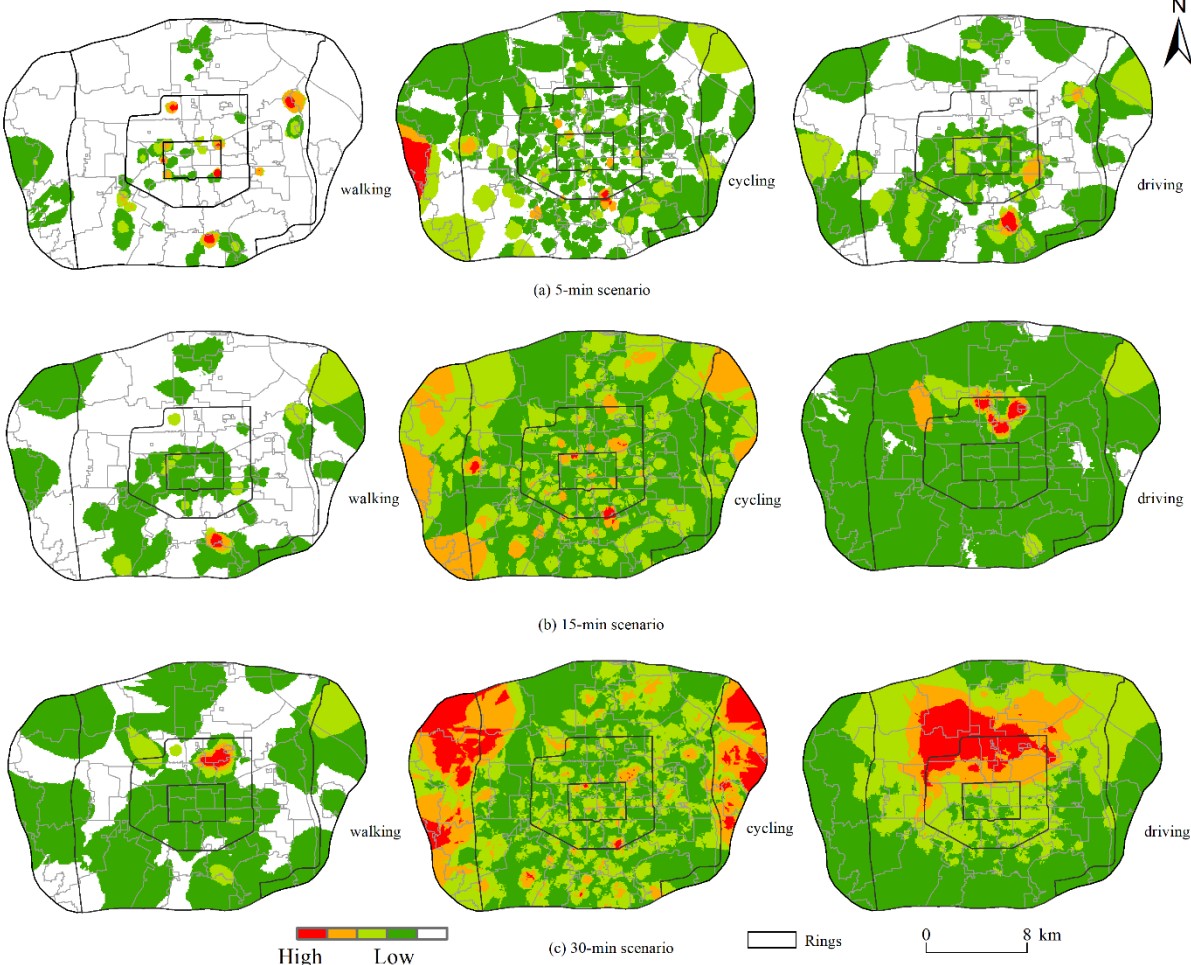

**Figure 4.** Spatial distribution of accessibility of three travel modes under different time thresholds.

Under the travel threshold of 5-min, the high value of accessibility is distributed around the ring road in walking mode. This is because Xi'an Park around city is situated along the city walls, which it is convenient and attractive to local inhabitants, transforming it into a favorable leisure and entertainment destination. High values of cycling and driving accessibility appear near Qujiang district. In recent years, Qujiang District has emphasized the development of cultural and tourism industries, resulting in the establishment of numerous parks with characteristics. These parks have comprehensive surrounding infrastructure, convenient transportation, thus strong attraction for residents. In addition, because the Qujiang District is predominantly a villa district with a small population, it is a high value area. From a 5-min to 30-min scenario, the high values of walking and driving accessibility have shifted from south to north, and the areas of Daming Palace Heritage Park and Baqiao Ecological Wetland Park have developed into new high-value center. Access to high values is evenly distributed in cycling modes, but as the time threshold increases, high-value gathering spots arise around the Third Ring Road's side. This is

because there are numerous comprehensive parks on the east of the East Third Ring Road, and the populations in this area which means that demand is low. On the whole, the overall accessibility level within the Ring Expressway is reasonably good. Nonetheless, the southwest area is a low-value area. In comparison, riding accessibility is best in the 15-min scenario.

### 3.5. Spatial Equity Analysis

Accessibility is merely a spatial manifestation of the allocation of public service resources, while scholars and policy makers focus on the underlying spatial equity [39]. The degree of curvature of the Lorenz curve can indicate the equity of resource allocation. Figure 4 demonstrates the inequitable distribution of urban park resources (Figure 5). A total of 10% of the population enjoys only 0.2% of urban park resources, whereas 20% enjoy 1.04%.

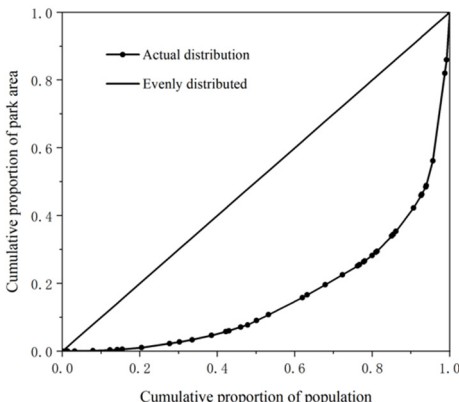

**Figure 5.** Lorentz curve of urban park resource allocation.

In order to investigate the spatial differentiation of equity of urban park resources allocation, this study calculates the location quotient of each street and categorizes the results into six groups (Figure 6, Table 4). In terms of the loop line, the most equitable route is located around the ring road. Additionally, there is significant disparity in the distribution of urban park resources between the second ring road and the third ring road. There are 16 streets that lack park distribution, which means that residents must cross roadways to use parks on daily basis. Due to the scarcity and small size of the parks, there are 16 streets with location quotient less than 0.2. The location quotient of 15 streets is above average. The greatest location quotient is in Baqiao Street which is ascribed to the street's national AAAA-level tourism attraction, Chanba National Wetland Park. Areas with large parks allow for high levels of urban park resource allocation per capita.

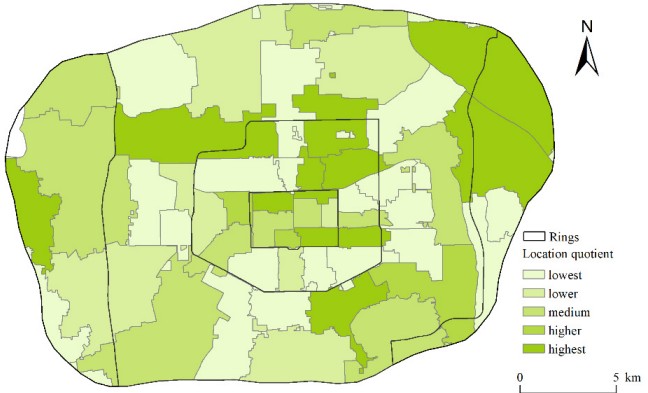

**Figure 6.** Street location quotient classification map.

**Table 4.** Street location quotient level table.

| Grade | Location Quotient | Number of Streets | Ratio |
|---|---|---|---|
| Lowest | <0.2 | 17 | 34.6% |
| Lower | 0.2–0.6 | 7 | 14.3% |
| Medium | 0.6–1.0 | 10 | 20.4% |
| Higher | 1.0–1.4 | 2 | 4.1% |
| Highest | >1.4 | 13 | 26.5% |

## 4. Discussion

### 4.1. Methodological Contributions

Accessibility is influenced by both geographic and non-spatial variables, residents' preferences for parks may be altered by non-spatial factors and traffic pat-terns [22]. However, G2SFCA only evaluates the ratio between the supply and the demand, and the majority of studies focus on park accessibility via a single-mode [21]. Incorporating non-spatial criteria into the accessibility assessment of parks has therefore been enhanced by this study. The contributions of this paper are most evident in the following aspects: Firstly, the park level and POI are incorporated in the analysis of the attraction coefficient. Compared to the G2SFCA, the SD and mean of iG2SFCA are smaller, and the transition between high and low values is more natural, resulting in more realistic findings. In addition, all parameters are designed to be flexible. The park weight can be adjusted according to the change of facility type. The type of POI can be modified to consider other factors, such as adding the number of parking lots or giving higher weight to specific facilities. Therefore, this methodology can be used as a computational model for assessing the accessibility of other amenities, such as schools, medical centers, and shopping malls. Additionally, prior research indicates that assessing the accessibility of a park based just on supply and demand is not a reliable strategy [10,25]. Secondly, it is integrated with big data to collect population data, compensating for the inadequacy of statistics [30]. This research uses Baidu Maps API to determine the travel time between residential neighborhoods and parks, and incorporates it into an accessibility model that can reflect the real travel behavior of residents [15,34]. Different time thresholds have a considerable impact on the accessibility distribution features, as demonstrated by the findings. Thirdly, this research employs location quotients to identify locations that may be underserved by parks, thereby assisting policy-makers in optimizing the layout of urban parks. Although the focus of this study is on parks, the methodology can be used as a computational model for assessing the accessibility of other public amenities.

### 4.2. Implications for Urban Park Planning

The distribution of spatial equity is similar to the spatial distribution of 15-min walking accessibility, demonstrating that accessibility can represent spatial equity to a certain extent [8]. According to the research results of accessibility and spatial equity, combined with the actual condition of Xi'an, optimization ideas are offered to help improve unbalanced park services as much as possible: the low accessibility and poor spatial equity streets are mostly distributed between Ring Road and the Second Ring Road, limited by high-density development model and land resources, turning irregular or small vacant lots into mini-parks or pocket parks [41] can increase the supply of park and coordinate the relationship between nature and urban development, and create a high-quality road landscape that is both ecologically and culturally sensitive. Additionally, Xi'an has a large number of heritage parks. The characteristics of heritage parks should be fully exploited to create a comprehensive park that incorporates natural scenery, history, and culture. For strip park, existing ecological corridors should be utilized to maintain the natural wetland, divide multi-functional areas, and improve the park layout. Moreover, the attractiveness of the park is strongly influenced by non-spatial variables. Therefore, public service facilities (such as restaurants, cafes, gymnasiums, and other recreational areas) might be added to the park's vicinity to increase its appeal to the citizens [22].

### 4.3. Limitations and Future Research

This study also has certain drawbacks. Firstly, we generate the OD matrix using centroid coordinates of the park as end point, but, people enter the park when they arrive at the entrance [36]. In the subsequent study, the OD matrix can be generated by acquiring the coordinates of the park entrance as the end point of the trip.

In addition to distance and park attraction, individual socio-economic variations play a significant role in determining individuals' inclination to visit a park [42], this article makes no distinction between individuals' travel inclinations (e.g., income, ethnoracial characteristics, age, gender, and disability). Recent research has shown that people with low socioeconomic status have less access to parks [43,44]. However, few research has examined the disparities in park accessibility across Xi'an's various demographic groups. Consequently, future research might categories the population to investigate the environmental justice of the park in greater depth.

Thirdly, the accessibility of park during working days and on weekends also deserves further discussing. People have various preferences for parks on weekdays and weekends [45]. The prevalence of weekends encourages the practice of long-distance travel. This study focuses solely on the impact of various travel modes on park accessibility. Future researchers should describe accessibility factors at various times to reinforce the findings.

### 5. Conclusions

At the street level, the distribution of park supply and population demand in Xi'an 3 city is uneven. Zhangbagou Street Park has the largest number, including 14 parks, although there are still nine streets without parks. The relationship between population geographical distribution and policies is strong. The ring road has a small population density. However, the high-tech zone is densely populated due to company development policy support.

The G2SFCA underestimates regional accessibility. The iG2SFCA can provide a more realistic evaluation, yet the spatial pattern of accessibility under both models is comparable. Specifically, a very accessible gathering area is located next to the park. As distance increases, the level of accessibility eventually declines.

Under three modes of transport, the reachable range expands as the travel duration increases. In walking mode, the reachable area extends from the core of the Ring Road, Tang Great Wall Heritage Park and Qujiangchi Heritage Park area. In the 5-min scenario, the area of riding accessibility is the largest. The average values of park accessibility and SD decline as time passes. At 30-min, the park service gap between communities is at its smallest, but some areas remain unreachable in walking mode.

High accessibility areas differ with travel mode. In walking and driving modes, high-accessibility area moves northward. In riding mode, high value areas of accessibility are located on both sides of the east and west third ring roads. Under the 15-min walking condition, high values areas appear near Huancheng Park, Qujiangchi Heritage Park, and Daming Palace Heritage Park, which corresponds to Huang's research findings [46].

The distribution of park resources in the study area is inequitable. Due to the fact that 14.5% of the streets are not distributed with parks, the residents' demand for parks cannot be addressed within the streets. 33.3% of the streets have a location quotient of less than 0.2, while 31.2% having a location quotient of greater than 1, indicating those streets have a higher level of equality.

**Author Contributions:** Methodology, Y.L.; Data curation, Y.L. and S.S.; Writing—original draft preparation, Y.L.; writing—review and editing, Y.X. and L.H.; Visualization, Y.L. All authors have read and agreed to the published version of the manuscript.

**Funding:** This research received no external funding.

**Data Availability Statement:** The dataset utilized and/or analyzed during the present study are available on reasonable request from the corresponding author.

**Conflicts of Interest:** The authors declare no conflict of interest.

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
