# Peer review of "Evaluation of Park Accessibility Based on Improved Gaussian Two-Step Floating Catchment Area Method: A Case Study of Xi’an City"

_buildings, doi:10.3390/buildings12070871_

Round 1

Reviewer 1 Report

The novelty of this paper is significant to publish on the Journal. Due to some issues that need improvement before a publication, my decision therefore is an acceptance with minor revisions.

Here are my comments for improving the manuscript:

·         Research approach and paper structure are good.

·         Please define abbreviations (e.g. POI. API, etc.) in the first use.

·         Some references are too old (over ten years). Please consider to update new ISI articles to improve this section.

Author Response

Point 1: Please define abbreviations (e.g. POI. API, etc.) in the first use.

Response 1: We added definition in the first use and used abbreviations in the following description

Point 2: Some references are too old (over ten years). Please consider to update new ISI articles to improve this section.

Response 2: We have updated the references. And it is necessary to quote some earlier literature, e.g. Hansen(1959), Yu(1999)

Please refer to the attachment for the  revised paper.

Reviewer 2 Report

Evaluation of park accessibility based on improved Gaussian 2 two-step floating catchment area method: A case study of Xi'an 3 City

 A brief summary

 The research works concentrated on how accessibility of selected park that plays a critical role in evaluating the quality of park construction. The paper proposed an improved Gaussian-based two-step floating catchment area method. Park accessibility in 15 the 5-min, 15-min, and 30-min scenarios have been measured. The method provides a new idea for the accessibility measurement of public facilities.  The results of the study provide theoretical justification for improving the park layout in Xi’an 3 City as considered case study.

Comments

Strengths of the paper:

  1. Quite well laid out paper, comprehensive, clear and in terms of descriptive content (apart from pretty awkward abbreviations like (iG2SFCA, G2SFCA, 2SFCA … – are they justified?)

Weakness of the paper:

1.      The abstract is too general and does not indicate what was achieved as a result of such extensive and systematic literature review

  1. Paper content and research works limited to review of literature different approaches and their description and comments; even then no all quoted references have been reviewed and discussed

3.      No clear and typological indication of research sample; it is rather a case study developed

  1. Research methodology presented in the Section 2.3 an the formulas 3 - 5 are own inventions / transformation of authors or do they rely on the other works / literature?  
  2. The model has not been verified by any method of research
  3. Discussion are not thoroughly developed in terms of considered case study application / extrapolation to other similar / different cases - section 4 must be significantly developed
  4. Conclusions (only 3!) presented in the section 5 are extremally limited and not trustworthy at all as focused to the researched case study – solid, practical, conclusions must proof what was achieved in so impressive study
  5. The overall merit of the paper is quite doubtful with opinion that the presented recognition on the evaluation of park accessibility by two (standard and improved) methods is not very much novel to the state of the art

Author Response

Point 1: pretty awkward abbreviations like (iG2SFCA, G2SFCA, 2SFCA … – are they justified?

Response 1: We added definition in the first use. These abbreviations refer to previous studies. E.g. Li (2021), Xing (2020), Wang (2018), Shalini(2016)

Point 2: The abstract is too general and does not indicate what was achieved as a result of such extensive and systematic literature review

Response 2: We have modified the abstract. The modification results are as follows:

Park accessibility plays a critical role in evaluating the quality of park construction. However, the conventional accessibility model ignores non-spatial factors, so it is crucial to use more complex methods for evaluating park accessibility. This study aims to establish an improved Gaussi-an-based two-step floating catchment area method (iG2SFCA) based on Point of Interest (POI), population data and Baidu map, to measure the park accessibility of various travel modes (walking, riding and driving modes) in 5-minute, 15 minute and 30 minute scenarios, and location quotient is used to assess spatial equity of parks. The results show that: (1) there are clear dis-parities between park supply and population demand at the street level (2) iG2SFCA evaluates the level and attractiveness of the park comprehensively. It is more sensitive to identifying accessi-bility, which can lead to a more realistic assessment of Park accessibility; (3) under the three modes of transportation, the accessible area of the park increases with time, and the accessibility difference between residential areas is the smallest under the 30 minute scenario. Overall, acces-sibility of park is relatively high, however there is an obvious tendency for the accessibility level decreases from the park as the center. The areas with poor accessibility appear in the north and southeast of the research area;(4) There are regional variances in the spatial equity of parks within Xi'an 3 City, and the park configuration needs to be optimized. These findings can provide the-oretical support for further optimizing the layout of park in Xi'an in order to improve the spatial equity of urban parks.

Point 3: Paper content and research works limited to review of literature different approaches and their description and comments; even then no all quoted references have been reviewed and discussed

Response 3: We have revised the introduction and discussion.

Point 4: No clear and typological indication of research sample; it is rather a case study developed

Response 4: The research sample of this paper is Xi'an parks, and the parks are classified.

Point 5: Research methodology presented in the Section 2.3 an the formulas 3 - 5 are own inventions / transformation of authors or do they rely on the other works / literature?

Response 5: G2SFCA is presented by Dai (2011) has been explained in the paper. iG2SFCA is based on G2SFCA. By referring to some research results, the main improvements of iG2SFCA include the following aspects: 1) consider the park grade; 2) join the influence of surrounding facilities (POI); 3) use time threshold instead of distance threshold.

We modified the section 2.3.2:

Although the G2SFCA assesses park accessibility from both supply and demand perspectives, there are still deficiencies. This paper is enhanced in two aspects as fol-lows.

Firstly, consider the contrasts in park quality and park attractiveness factors. In addition to spatial features, park size and nearby facilities also have a significant effect on park accessibility [10,36,37]. As a result, we assign attraction weights of 0.6, 0.4, and 0.2 to the three levels of classification. With an appropriate walking distance of 800 meters [38], we take the normalized quantity of POI within the 800-meter buffer outside the park as an indicator of attractiveness. The normalized quantity is calculated as the attractiveness factor. This article focuses on five types of POIs that are closely related to leisure and entertainment [39]: catering services, shopping services, sports and leisure services, scenic places, and scientific, educational, and cultural services; with a total of 77,089 POIs retrieved.

Secondly, the distance threshold is replaced with the time threshold. Due to the advancement of big data technology, it is possible to utilize a Baidu map to acquire travel time under multiple travel modes, which facilitates the analysis of the park’s spatial accessibility and reveals the accessibility variations under different traffic modes [10]. Consequently, this study used three modes of transportation - walking, riding, and driving - to examine the accessibility change characteristics at various time thresholds.

Point 6: The model has not been verified by any method of research

Response 6: We refer to the relevant literature on the improvement of G2SFCA and find that there is no index to judge the improvement effect (Xing, 2020; Hu, 2020; Xing, 2018; Langford, 2016). The improvement effect is usually illustrated the improvement by comparing the space and quantity of the two methods.

Point 7: Discussion are not thoroughly developed in terms of considered case study application / extrapolation to other similar / different cases - section 4 must be significantly developed

Response 7: We have amended the discussion. The modification results are as follows:

4.1. Methodological Contributions

Accessibility is influenced by both geographic and non-spatial variables, residents’ preferences for parks may be altered by non-spatial factors and traffic pat-terns [22]. However, G2SFCA only evaluates the ratio between the supply and the demand, and the majority of studies focus on park accessibility via a single-mode [21]. Incorporating non-spatial criteria into the accessibility assessment of parks has there-fore been en-hanced by this study. The contributions of this paper are most evident in the following aspects: Firstly, the park level and POI are incorporated in the analysis of the attraction coefficient. Compared to the G2SFCA, the SD and mean of iG2SFCA are smaller, and the transition between high and low values is more natural, resulting in more realistic findings. Additionally, prior research indicates that assessing the accessibility of a park based just on supply and demand is not a reliable strategy [10,25]. Secondly, it is inte-grated with big data to collect population data, compensating for the inadequacy of statistics [30]. This research uses Baidu Maps API to determine the travel time between residential neighborhoods and parks, and incorporates it into an accessibility model that can reflect the real travel behavior of residents [15,34]. Different time thresholds have a considerable impact on the accessibility distribution features, as demonstrated by the findings. Thirdly, this research employs location quotient to identify locations that may be underserved by parks, thereby assisting policy-makers in optimizing the layout of urban parks. Although the focus of this study is on parks, the methodology can be used as a computational model for assessing the accessibility of other public amenities.

4.2. Implications for Urban Park Planning

The distribution of spatial equity is similar to the spatial distribution of 15-min walking accessibility, demonstrating that accessibility can represent spatial equity to a certain extent [8]. According to the research results of accessibility and spatial equity, combined with the actual condition of Xi’an, optimization ideas are offered to help improve unbalanced park services as much as possible : the low accessibility and poor spatial equity streets are mostly distributed between Ring Road and the Second Ring Road, limited by high-density development model and land resources, turning irregu-lar or small vacant lots into mini-parks or pocket parks [41] can increase the supply of park and coordinate the relationship between nature and urban development, and cre-ate a high-quality road landscape that is both ecologically and culturally sensitive. Ad-ditionally, Xi’an has a large number of heritage parks. The characteristics of heritage parks should be fully exploited to create a comprehensive park that incorporates natu-ral scenery, history, and culture. For strip park, existing ecological corridors should be utilized to maintain the natural wetland, divide multi-functional areas, and improve the park layout. Moreover, the attractiveness of the park is strongly influenced by non-spatial variables. Therefore, public service facilities (such as restaurants, cafes, gymnasiums, and other recreational areas) might be added to the park’s vicinity to in-crease its appeal to the citizens [22].

4.3. Limitations and Future Research

This study also has certain drawbacks. Firstly, we generate the OD matrix using centroid coordinates of the park as end point, but, people enter the park when they ar-rive at the entrance [36]. In the subsequent study, the OD matrix can be generated by acquiring the coordinates of the park entrance as the end point of the trip.

Additionally, in addition to distance and park attraction, individual so-cio-economic variations play a significant role in determining individuals’ inclination to visit a park [42], this article makes no distinction between individuals’ travel incli-nations (e.g., income, ethno-racial characteristics, age, gender, and disability). Recent research has shown that people with low socioeconomic status have less access to parks [43,44]. However, few research has examined the disparities in park accessibility across Xi'an’s various demographic groups. Consequently, future research might cate-gories the population to investigate the environmental justice of the park in greater depth.

Thirdly, the accessibility of park during working days and on weekends also deserves further discussing. People have various preferences for parks on weekdays and weekends [45]. The prevalence of weekends encourages the practice of long-distance travel. This study focuses solely on the impact of various travel modes on park accessibility. Future re-searchers should describe accessibility factors at vari-ous times to reinforce the findings.

Point 8: Conclusions (only 3!) presented in the section 5 are extremally limited and not trustworthy at all as focused to the researched case study – solid, practical, conclusions must proof what was achieved in so impressive study

Response 8: We summarized the research results and improved the conclusion.

At the street level, the distribution of park supply and population demand in Xi'an 3 city is uneven. Zhangbagou Street Park has the largest number, including 14 parks, although there are still 9 streets without parks. The relationship between population geographical distribution and policies is strong. The ring road has a small population density. However, the high-tech zone is densely populated due to company devel-op-ment policy support.

The G2SFCA underestimates regional accessibility. The iG2SFCA can provide a more realistic evaluation, yet the spatial pattern of accessibility under both models is comparable. Specifically, a very accessible gathering area is located next to the park. As distance increases, the level of accessibility eventually declines.

Under three modes of transport, the reachable range expands as the travel duration increases. In walking mode, the reachable area extends from the core of the Ring Road, Tang Great Wall Heritage Park and Qujiangchi Heritage Park area. In 5-min scenario, the area of riding accessibility is the largest. The average values of park accessibility and SD decline as time passes. At 30 minutes, the park service gap between communities is at its smallest, but some areas remain unreachable in walking mode.

High accessibility areas differ with travel mode. In walking and driving modes, high-accessibility area moves northward. In riding mode, high value areas of accessi-bility are located on both sides of the east and west third ring roads. Under the 15-minute walking condition, high values areas appear near Huancheng Park, Qujiangchi Heritage Park, and Daming Palace Heritage Park, which corresponds to Huang’s research findings [46] .

The distribution of park resources in the study area is inequitable. Due to the fact that 14.5% of the streets are not distributed with parks, the residents’ demand for parks cannot be addressed within the streets. 33.3% of the streets have a location quotient of less than 0.2, while 31.2% having a location quotient of greater than 1, indicating those streets have a higher level of equality.

Point 9: The overall merit of the paper is quite doubtful with opinion that the presented recognition on the evaluation of park accessibility by two (standard and improved) methods is not very much novel to the state of the art

Response 9: The value of this paper lies in the introduction of non-spatial factors (facilities around the park) into the calculation of accessibility, which may not be novel enough, but we think this is a further exploration of the calculation method of accessibility.

Please refer to the attachment for the  revised paper.

Round 2

Reviewer 2 Report

Evaluation of park accessibility based on improved Gaussian 2 two-step floating catchment area method: A case study of Xi'an 3 City – rev.2

The Authors thoroughly thought out my comments and concerns therefore significantly improved their revised paper accordingly.

However, the comment 3 (No clear and typological indication of research sample; it is rather a case study developed) answered “The research sample of this paper is Xi'an parks, and the parks are classified” it is still insufficient. Might the results be:

·       compared to other similar / different parks?

·       discussed with parks of the same size and near location?

·       generalized in terms of methods applied?

The Authors are encouraged to develop the issue and afterwards the manuscript may be published.

Author Response

Point 1: The Authors thoroughly thought out my comments and concerns therefore significantly improved their revised paper accordingly.

However, the comment 3 (No clear and typological indication of research sample; it is rather a case study developed) answered “The research sample of this paper is Xi'an parks, and the parks are classified” it is still insufficient. Might the results be:

  • compared to other similar / different parks?
  • discussed with parks of the same size and near location?
  • generalized in terms of methods applied?

Response 1: Thank you very much for your comments. In response to your questions, the following replies are provided.

Due to the limitations of data acquisition, we did not compare to other similar / different parks, but considering the universality of the method:

We grade the parks and select five types of POI to reflect the attractiveness of park, but all parameters are designed to be flexible. The park weight can be adjusted according to the change of facility type. The type of POI can be modified to consider other factors, such as adding the number of parking lots or giving higher weight to specific facilities. Therefore, this methodology can be used as a computational model for assessing the accessibility of other amenities, such as schools, medical centers and shopping centers.

Corresponding changes are made in the discussion section.